# Argument mining as a multi-hop generative machine reading comprehension task

**Boyang Liu[1], Viktor Schlegel[1,2], Riza Batista-Navarro[1], Sophia Ananiadou[1,3]**
[1] Department of Computer Science, The University of Manchester, UK
[2] ASUS Intelligent Cloud Services (AICS), Singapore
[3] Artificial Intelligence Research Center, AIST
{boyang.liu-2@postgrad., riza.batista@, sophia.ananiadou@ }manchester.ac.uk
viktor_schlegel@asus.com

## Abstract

Argument mining (AM) is a natural language processing task that aims to generate an argumentative graph given an unstructured argumentative text. An argumentative graph that consists of argumentative components and argumentative relations contains completed information of an argument and exhibits the logic of an argument. As the argument structure of an argumentative text can be regarded as an answer to a "why" question, the whole argument structure is therefore similar to the "chain of thought" concept, i.e., the sequence of ideas that lead to a specific conclusion for a given argument (Wei et al., 2022). For argumentative texts in the same specific genre, the "chain of thought" of such texts is usually similar, i.e., in a student essay, there is usually a major claim supported by several claims, and then a number of premises which are related to the claims are included (Eger et al., 2017). In this paper, we propose a new perspective which transfers the argument mining task into a multi-hop reading comprehension task, allowing the model to learn the argument structure as a "chain of thought". We perform a comprehensive evaluation of our approach on two AM benchmarks and find that we surpass state-of-the-art results.[1] A detailed analysis shows that specifically the "chain of thought" information is helpful for the argument mining task.

## 1 Introduction

Argument mining involves automatically identifying and extracting arguments and their relationships from natural language texts, such as persuasive essays or political speeches. This task aims to identify the various claims and supporting evidence used to support a particular argument.

AM can be broken down into four different subtasks (Eger et al., 2017): *argumentative component*

*segmentation* (**ACS**) aims to separate argumentative units from non-argumentative units; *argumentative component classification* (**ACC**) focuses on classifying argumentative components (AC) into specific types; *argumentative relation identification* (**ARI**) determines which ACs are related to each other, and the direction of these argumentative relations (AR); and *argumentative relation classification* (**ARC**) handles the labelling of the ARs. The ARI and ARC subtasks can be combined together as an *argumentative relation identification and classification* (**ARIC**) subtask where the model needs not only to predict the existence and the direction (if exists) of an AR but also its type (Si et al., 2022).

The argument structure of an argumentative text can be regarded as an answer to a "why" question; therefore, the whole argument structure is similar to the "chain of thought" concept—spelling out a sequence of reasoning steps (forming a reasoning path) that leads to a specific conclusion for a given argument (Wei et al., 2022). For argumentative texts of the same genre, the structure of such "chains of thought" tends to be similar: In a student essay, there is usually a major claim supported by several other claims, and then a number of premises which are related to the claims (Eger et al., 2017). Similarly, for scientific abstracts such as shown in Figure 1, premises about experimental results are used to support aspect-based claims (i.e., AC5 in Figure 1 mentions three aspects: postoperative IOP, bleb morphology and complications), which are further used to support a high-level claim (AC6 in Figure 1) as the conclusion of the abstract. However, most of the previous works (Mayer et al., 2020; Rodrigues and Branco, 2022; Saadat-Yazdi et al., 2023) ignore such structural similarity and mainly pay attention to single ACs for the AC-related subtasks and AC pairs for the AR-related subtasks.

To enable the model to learn such chains and

---

[1]Our code is availble from https://github.com/Boyang-L/MRC-GEN4AM

extract argument structure simultaneously, we propose to convert AM into a generative multi-hop machine reading comprehension (MRC) task (Yavuz et al., 2022). It is a sequence-to-sequence task where the input sequence is a combination of a query as well as a context, and the output sequence is the answer with the reasoning path showing how the model obtains the answer. Concretely, given an AC as a *query* and the whole text as the *context*, our approach predicts both the reasoning *path* and the type of the AC or any other ACs related to the query AC according to the different subtasks of argument mining as *answer*. Here, the reasoning path is a path from the ROOT AC to the query AC, where the ROOT AC is defined as the source node of the longest path in an argumentative graph, which usually contains the core opinion of an argument, such as the main conclusion of the abstract. An example is shown in Figure 1. It is worth mentioning that the direction of the path is reversed from the AR so that the model can learn the high-level semantics first. The reason that we chose the generative paradigm is so that only the output is changed and no extra parameters are included, thus letting the model leverage the path information. Therefore, we can directly test the impact of the path information.

Furthermore, to alleviate the bias arising from the order of answers learned by the generative model, we propose a two-direction-based method that allows the model to learn the output sequence from both directions.

To summarise, our contributions are as follows: 1) to our best knowledge, we are the first to cast the argument mining task as a multi-hop MRC task to let the model learn argumentative reasoning chains; 2) to alleviate the bias in the order of answers learned by the generative model, we propose a two-direction-based method; 3) extensive experimentation on two different benchmark datasets shows that our framework outperforms related models on most subtasks.

## 2  Related Work

**Argument Mining.** Recently, AM has received increasing attention from the research community, and a range of models has been proposed (Eger et al., 2017; Gemechu and Reed, 2019; Hewett et al., 2019; Lytos et al., 2019; Dutta et al., 2020; Liu et al., 2022; Cheng et al., 2022; Liu et al., 2023). Most of them mainly pay attention to local features

to solve the argument mining task. For example, some models only leverage single ACs or a pair of ACs as input without wider contextual information (Mayer et al., 2020; Accuosto et al., 2021). Despite using similar input, Galassi et al. (2023) recognised that individual propositions may not always provide enough information for AM.

In contrast, alternative studies take advantage of a broader document context by providing the entire text to their models. They adopt different approaches for the AM task, such as treating it as a sequence tagging task (Eger et al., 2017), a dependency parsing task (Ye and Teufel, 2021) or employing a prompt-based model (Dutta et al., 2022).

Further, some researchers found that combining plain text with its structural information is beneficial for AM. Examples include the distance between two ACs (Galassi et al., 2023); the position of the AC in the document or paragraph (Potash et al., 2017; Bao et al., 2021); discourse parser features (Hewett et al., 2019); and argumentative zoning information (Liu et al., 2022). Although these types of information are helpful for the model to understand the structure of an argumentative text, a model cannot understand the "chain of thought" from such structural information and the plain text only.

**Other tasks as MRC task.** A recent successful trend involves treating various other natural language processing tasks as an MRC task, such as relation extraction (Levy et al., 2017), named entity recognition (Li et al., 2019), event extraction (Liu et al., 2020) and coreference resolution (Wu et al., 2020). Unlike traditional MRC, multi-hop MRC requires the machine to not only predict the answer of a given query, but also provide the reasoning path showing how the answer is obtained. In this paper, we propose a model to transfer the argument mining task into a multi-hop MRC task so that the model can learn the "chain of thought" from the prediction of the reasoning path.

## 3  Method

As previously mentioned, we assume that the ACS subtask has been solved and we only focus on the other subtasks. There are two ways to combine the remaining subtasks, i.e., ACC + ARIC and ACC + ARI + ARC. Considering that there is no established preference in existing literature—the former is used on the AbstRCT dataset (Mayer et al., 2020; Si et al., 2022; Galassi et al., 2023) and the latter

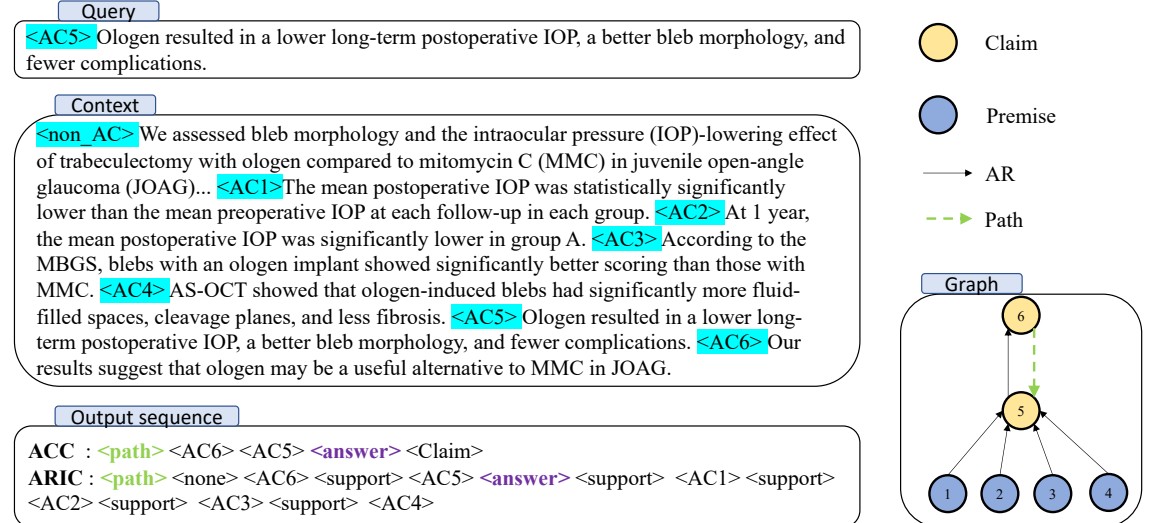

Figure 1: An example of how our model works on the AbstRCT dataset. Given AC5 as a query and the whole abstract as the context, the output sequence is the combination of the predicted path (the tokens between the special tokens <path> and <answer>) and the answer (the tokens following <answer>). In this example, the path contains two ACs, the R00T node AC6 and the query node AC5. The answer varies depending on the subtask, i.e., for the ACC subtask, the answer is <Claim> which means the type of AC5 is claim; for the ARIC subtask, the answer means that AC1, AC2, AC3 and AC4 are ACs that support AC5. The whole argumentative graph of the context in the right part can be obtained after all ACs are used as queries. All relation types in this graph are "support".

is used on the PE dataset (Eger et al., 2017; Kurib-ayashi et al., 2019; Bao et al., 2021)—we focus on all four subtasks to enable a fair comparison.

In this section, we introduce how to cast argument mining as a multi-hop generative MRC task, which consists of four parts, namely, *<query, context, answer, path>*, where the input sequence is the query and the context, and the output sequence is the answer and the path. We introduce the details of these two sequences in the following.

## 3.1 Input sequence

The input of the model is a word sequence that contains the *context* and the *query*.

**Context Representation.** Let $DOC = \{t_1, t_2, ..., t_l\}$ denote an argumentative document, where $t_i$ represents the $i$-th token in $DOC$. As mentioned in Section 1, we assume that the position of all ACs in the document is known. The context representation consists of all tokens in the document, with additional tokens inserted to denote the boundaries of each AC. Specifically, we insert an $< AC\_i >$ token before the start token of the $i$-th AC token sequence in document $D$ and a $< non\_AC >$ token before the first token of all non-argumentative sequences. The context sequence $C$ is shown below:

$$C =< AC\_1 > t_1, t_2... < non\_AC > ...t_i...$$
(1)

**Query Representation.** We use the AC tokens as queries. There are two types of query representation $Q$ according to the subtasks: a unary query $Q_u$ that consists of one AC for the ACC, ARI and ARIC subtasks, and a binary query $Q_b$ which is composed of two ACs separated by the special tokens for the ARC subtask:

$$Q_u = t_{AC_{i_1}}, ..., t_{AC_{i_n}}$$
$$Q_b =< AC_i > t_{AC_{i_1}}, ..., t_{AC_{i_n}}$$
$$< AC_j > t_{AC_{j_1}}, ..., t_{AC_{j_n}}$$
(2)

Finally, the input sequence is the concatenation of the query $Q$ and the context $C$ differentiated by two specific tokens <query> and <context>.

$$Input =< query > Q < context > C$$
(3)

## 3.2 Output sequence

The output sequence of a given query consists of two parts, the *answer* sequence and the *path* sequence.

**Answer Representation.** The answer representation differs for different subtasks. The details are

| | Path Representation | Answer Representation |
|---|---|---|
| ACC | $< AC_{p_1} > .. < AC_{p_n} >$ | $< ACT_i >$ |
| ARI | $< AC_{p_1} > .. < AC_{p_n} >$ | $< AC_{a_1} > ... < AC_{a_n} >$ |
| ARC | $< AC_{p_1} > .. < AC_{p_n} >$ | $< ART_i >$ |
| ARIC | $< ART_1 >< AC_{p_1} > ... < ART_n ><$ $AC_{p_n} >$ | $< ART_1 >< AC_{a_1} > ... < ART_n ><$ $AC_{a_n} >$ |

Table 1: The path and answer representations for different subtasks. Here, $< AC_{p_1} >$ is the ROOT AC and $< AC_{p_n} >$ denotes the query AC; $< AC_{a_1} > ... < AC_{a_n} >$ are ACs that point to the query AC; $< ACT_i >$ and $< ART_i >$ represent the AC type and the AR type.

shown in Table 1. For the ACC subtask, given an AC as a query, the model needs to predict the type of the AC. Similar to the ACC subtask, given a pair of ACs with a relation between them, the answer for the ARC subtask is a specific token $< ART_i >$ representing the AR type. For the ARI subtask, the answer to a given query contains all ACs that are related to it. As for the ARIC subtask, given an AC as a query, the model needs to predict all the ACs related to it and the relation types at the same time. Therefore, the ARIC subtask on the AbstRCT dataset is defined as a directed relation classification task with three possible types: None, Support and Attack. Here, a true positive is an outcome where the model correctly predicts both the relation type and direction, given two ACs. Therefore, the answer contains both the type tokens $< ART_i >$ and the AC tokens $< AC_{a_i} >$.

**Path Representation.** A reasoning path starts from the ROOT node and ends at the query AC. We define $< AC_i >$ as ROOT when it satisfies the following: first, $< AC_i >$ does not point to any other $< AC_j >$; second, there is a path that starts with $< AC_i >$ and the length of the path is the longest among all the paths. We use the path representation without relation types for the ACC, ARI and ARC subtasks. For the ARIC subtask, we also include type information in the path representation to align it with the answer representations. The path representations are shown in Table 1.

The final output sequence is the combination of the path sequence and the answer sequence denoted by two specific tokens <path> and <answer>.[2]

$$Output = < path > P < answer > A \quad (4)$$

---
[2]Incorporating the whole graph rather than the path information might incorporate additional global information, but it performed worse in initial experiments (see Appendix B).

### 3.3 Output Order Debiasing

For the ARI and ARIC subtasks, the output is a set of correct answers rather than a single one, which raises the question of its order in the output sequence. Arranging them in ascending order ($< AC_1 >, < AC_2 >, ...$) might teach the model an order bias and disincentivise the generation of $< AC_j >$ tokens when an $< AC_{i>j} >$ token is generated, resulting in unrecoverable errors.

To alleviate this issue, we propose a simple but effective *two-direction* augmentation method. Concretely, for each ARI and ARIC query, we create two training samples with different answers, one sorted by the ordinal number of the AC tokens appearing in the sequence and the other one sorted in reverse.

### 3.4 Training and Inference

**Training.** The output sequence contains two parts, path and answer; this is a form of a multi-task paradigm that includes the learning of both sequences jointly. Thus, during training, we calculate the loss of each part separately and then sum the two as our final loss function.

$$\mathcal{L} = (1 - \lambda)\mathcal{L}_{path} + \lambda\mathcal{L}_{answer} \quad (5)$$

where $\mathcal{L}$ is the cross entropy loss function and $\lambda \in [0, 1]$ denotes the weights of $\mathcal{L}_{answer}$.

To let the model leverage the information from pre-trained language models, we employ a warm start strategy inspired by (Guo et al., 2022). To be specific, we use the embedding of number $i$ as the initial representation for the specific token $< AC_i >$ instead of training the embedding from scratch. The model will cost more time and the training will be unstable if the model learns the embeddings from scratch during fine-tuning. The representation of the context is kept the same for all four subtasks.

During the training phase, we optimise the negative loglikelihood using teacher forcing.[3]

**Inference.** During inference, we use beam search decoding to obtain the output sequence $Output$ in an autoregressive manner. We then post-process the decoded sequence using the answer indicator (<answer>) to obtain the answer and convert the output sequence into labels according to their meanings described in Section 3.2.

# 4 Empirical study

## 4.1 Dataset

We used two publicly available datasets to evaluate our model and compare it with results obtained by previously proposed models. Descriptive information on these two datasets is provided in Table 2. Next, we will give a brief description of them.

|  | Documents | All ACs | All ARs |
|---|---|---|---|
| Neo_train | 350 | 2267 | 1427 |
| Neo_dev | 50 | 326 | 210 |
| Neo_test | 100 | 686 | 424 |
| Gla_test | 100 | 594 | 367 |
| Mix_test | 100 | 570 | 329 |
| PE | 2235 | 6095 | 3832 |

Table 2: Statistics of datasets used in our paper. In order to show the difference between different test sets of the AbstRCT dataset, we report the data statistics of three test sets separately. Here, *Neo*, *Gla* and *Mix* represent neoplasm, glaucoma and mixed. The statistics of the PE dataset are paragraph level.

**AbstRCT** (Mayer et al., 2020) is comprised of 659 abstracts from biomedical publications. These abstracts are annotated with three types of ACs (*major claim*, *claim* and *evidence*) and two types of ARs (*support* and *attack*). The dataset is divided into three parts. The first part, the neoplasm corpus, is further divided into training, development and test sets. Additionally, there are two separate test sets: one for glaucoma and another for a mix of topics. The argumentative graphs in this dataset exhibit a non-tree (graph) structure.

**Persuasive Essays (PE)** (Eger et al., 2017) contains of 402 essays (2235 paragraphs). Three types of ACs (*MajorClaim*, *Claim*, and *Premise*) are used for annotation. In addition, there are two types of

ARs: *support* and *attack*. In order to maintain consistency with previous studies (Bao et al., 2021; Kuribayashi et al., 2019), each paragraph with its ACs and ARs is considered as an instance. Here, each AC is associated with *at most one* outgoing AR, which means that the argumentative graph of each paragraph has a tree or forest structure.

## 4.2 Evaluation and Implementation

**Evaluation.** For the AbstRCT dataset, we follow previous studies (Mayer et al., 2020; Si et al., 2022; Galassi et al., 2023) by merging *major claim* and *claim* into a single category. All results on both datasets are averaged scores of three different random seeds and are reported as macro-averaged F1 scores. For the AbstRCT dataset, we use the same train-development-test split as Si et al. (2022). For the PE dataset, we keep the same train-test split and randomly select 10% of the training set for validation, like Bao et al. (2021) do.

**Implementation.** We fine-tune a BART-Base model (Lewis et al., 2020) for the PE dataset and a BioBART-Base model (Yuan et al., 2022) for the AbstRCT dataset. Regarding the learning rate, we set it to 3e-5 for the ACC and ARC subtasks, 2e-5 for the ARI subtask, and 8e-5 for the ARIC subtask. The max sequence length is 512 for the PE dataset and 768 for the AbstRCT dataset. The batch size is 16, and we assign a value of 0.7 to the hyperparameter $\lambda$. During inference, we employ beam search with a beam size of 4 for decoding purposes. To optimise our model, we employ AdamW (Loshchilov and Hutter, 2017). We train our model for 15 epochs except for the ARC subtask where we train it for 20 epochs and select the best checkpoint on the development set.

## 4.3 Baselines

We compare our method it with the following baselines on AbstRCT:

**ResArg** (Galassi et al., 2018) is a residual network model combined with a long short-term memory (LSTM) network that jointly addresses the ACC, ARI and ARIC subtasks.

**ResAttArg** (Galassi et al., 2023) is an extension of the ResArg model that includes an attention module and ensemble learning. Both ResArg and ResAttArg have an average and an ensemble version.

**SeqMT** (Si et al., 2022) implements a multi-task learning framework that leverages the sequential dependency between the ACC and ARIC subtasks

---

[3]Note that we experimented with multi-task learning by training the model on all subtasks at once, but this set-up performed worse than the formulation introduced above (see Appendix A for details).

|  | ACC | | | ARIC | | |
|---|---|---|---|---|---|---|
|  | Neo | Gla | Mix | Neo | Gla | Mix |
| ResArg(avg) | 86.18 | 85.53 | 86.74 | 59.15 | 57.23 | 60.31 |
| ResArg(Ensemble) | 86.38 | 87.13 | 87.59 | 63.16 | 61.86 | 68.35 |
| ResAttArg(avg) | 86.19 | 86.26 | 87.51 | 66.49 | 62.68 | 63.47 |
| ResAttArg(Ensemble) | 87.87 | 87.71 | 89.70 | 70.92 | 68.40 | 67.66 |
| SeqMT | 91.89 | 92.35 | 92.21 | 71.24 | 73.27 | 72.71 |
| MRC_GEN | **92.76*** | **92.62*** | **93.97*** | **74.97*** | **74.28*** | **73.87*** |

Table 3: Overall results on the AbstRCT dataset. Here, *Neo*, *Gla* and *Mix* correspond to the results achieved for the neoplasm, glaucoma and mixed test sets, respectively. The highest scores are emboldened. * indicates statistically significant improvements over the baselines with our model, according to pair-wise t-test with $p < 0.05$.

|  | ACC | | | | ARI | | | ARC | | |
|---|---|---|---|---|---|---|---|---|---|---|
|  | Macro | MC | Claim | Premise | Macro | Rel | No-Rel | Macro | Support | Attack |
| Joint-ILP | 82.6 | 89.1 | 68.2 | 90.3 | 75.1 | 58.5 | 91.8 | 68.0 | 94.7 | 41.3 |
| St-SVM-full | 77.6 | 78.2 | 64.5 | 90.2 | - | 60.1 | - | - | - | - |
| Joint-PN | 84.9 | 89.4 | 73.2 | 92.1 | 76.7 | 60.8 | 92.5 | - | - | - |
| Span-LSTM | 87.3 | - | - | - | 81.1 | - | - | 79.0 | 96.8 | **61.1** |
| BERT-Trans | 88.4 | 93.2 | 78.8 | 93.1 | 82.5 | 70.6 | 94.3 | **81.0** | - | - |
| MRC_GEN | **89.2*** | **94.8*** | **79.6*** | **93.2** | **82.7*** | **70.9*** | **94.4** | 78.2* | **97.7*** | 58.9* |

Table 4: Overall results on the PE dataset.* indicates statistically significant difference between the baselines with our model, according to pair-wise t-test with $p < 0.05$.

by transferring the representation of the input and output of the ACC subtask to the ARIC subtask.

For PE, we compare with the following baseline approaches:

**Joint-ILP** (Stab and Gurevych, 2017) is an end-to-end argumentation structure parser that globally optimizes ACs and ARs through Integer Linear Programming (ILP).

**St-SVM-full** (Niculae et al., 2017) is a linear structured SVM which formulates AM as inference in a full factor graph (Kschischang et al., 2001), which solves the ACC and ARI subtasks jointly.

**Joint-PN** (Potash et al., 2017) is a joint model based on a Pointer Network architecture to classify types of ACs and identify relations between ACs.

**Span-LSTM** (Kuribayashi et al., 2019) is an LSTM-minus-based span representation model with argumentative markers as extra information.

**BERT-Trans** (Bao et al., 2021) is a neural transition-based model designed for ACC and ARI subtasks. It is now the state-of-the-art (SOTA) model on this dataset.

## 4.4 Main Results

Performance comparisons between our model and the baselines are shown in Table 3 and Table 4. Our model achieves SOTA results on most of the tasks

for both datasets, even though our model is a fine-tuned BART-Base model, while other baselines are considerably more complex than ours.

Specifically, on the AbstRCT dataset, our model achieves the best performance on both the ACC and ARIC subtasks on all test sets. We also observe that for the PE dataset, our improvement is smaller on the ARI subtask and the performance does not reach that of the BERT-Trans model on the ARC subtask. However, BERT-Trans uses additional features, such as the relative distance between each two ACs and the bag-of-words vectors to improve the performance, while such features are not included in our model.

## 4.5 Ablation Study

We perform ablation experiments to investigate the effect of our method design on the overall performance on both benchmarks. There are mainly three models for the ablation study, MRC_GEN(-path) is designed to test whether the path information can improve the performance; MRC_GEN(-td) aims to test the effect of the two-direction method; and MRC_GEN(-ws) is for exploring the impact of warm start. Since the two-direction method is only suitable for the ARI and ARIC subtasks where the answer is a sequence of special tokens,

|  | ACC | | | ARIC | | |
|---|---|---|---|---|---|---|
|  | Neo | Gla | Mix | Neo | Gla | Mix |
| MRC_GEN | 92.76 | 92.62 | 93.97 | **74.97** | **74.28** | **73.87** |
| MRC_GEN(-path) | 92.29 | 92.27 | 93.72 | 72.47 | 70.10 | 71.20 |
| MRC_GEN(-td) | - | - | - | 74.29 | 71.90 | 73.50 |
| MRC_GEN(-ws) | **92.83** | **92.72** | **94.11** | 74.23 | 70.12 | 73.09 |

Table 5: Results of ablation experiments on the AbstRCT dataset. MRC_GEN(-path) denotes that the model only needs to predict the answer without the path information; MRC_GEN(-td) means that the two-direction method is excluded; MRC_GEN(-ws) uses a cold start method and the specific tokens are trained from scratch.

|  | ACC | | | | ARI | | | ARC | | |
|---|---|---|---|---|---|---|---|---|---|---|
|  | Macro | MC | Claim | Premise | Macro | Rel | No-Rel | Macro | Support | Attack |
| MRC_GEN | **89.2** | **94.8** | **79.6** | **93.2** | **82.7** | **70.9** | **94.4** | **78.2** | **97.7** | **58.9** |
| MRC_GEN(-path) | 88.0 | 93.3 | 78.2 | 92.7 | 79.8 | 65.6 | 93.8 | 75.7 | 97.4 | 54.1 |
| MRC_GEN(-td) | - | - | - | - | 81.7 | 69.0 | 94.4 | - | - | - |
| MRC_GEN(-ws) | 87.9 | 93.5 | 78.0 | 92.5 | 81.1 | 68.6 | 93.5 | 74.0 | 97.5 | 50.5 |

Table 6: Results of ablation experiments on the PE dataset.

MRC_GEN(-td) can not be applied to the ACC subtask. The results are shown in Table 5 and Table 6.

**Two-Direction Method.** Comparing the results of MRC_GEN(-td) and MRC_GEN, reveals that our two-direction method works for the ARI and ARIC subtasks. To explain this observation, we calculate the percentage of examples where the order of answer AC tokens is strictly ascending. For the ARIC subtask on the AbstRCT dataset, the percentage is 27.09%. The proportion on the PE dataset for the ARI subtask is 34.83%. This shows that forcing the model to learn only from examples in ascending order may inhibit its ability to generate the correct answer.

**Warm Start.** The warm start method improves the performance in most cases, in line with literature (Guo et al., 2022). Leveraging pre-trained embeddings as starting points for newly-added tokens is better than training their embeddings from scratch, since the size of the dataset for fine-tuning is much smaller than that of the pre-training dataset. Therefore, it is difficult for the model to fully learn the semantics of the new tokens only from the fine-tuning data. However, from Table 5, we also find that warm start is clearly hurting the ACC task on AbstRCT. From an intuitive point of view, each AC token needs to include two types of information: the location of the AC and the content of the AC. The warm start method only includes location information since we use the embedding of the numbers as a starting point for each AC token, while the model also needs to learn that this token

is about the content of the AC. However, overlearning the representation from the warm start method may also decrease the performance. Therefore, good hyper-parameters are also important to let the model learn a balanced representation of these two types of information. Thus, we believe that the hyper-parameters for the ACC subtask on the PE dataset are good enough to learn a balanced representation.

**Path Information.** In general, without path information, the scores drop for all tasks on both datasets. The decrease is more obvious on the relation-based tasks such as ARI, ARC and ARIC as opposed to ACC. Specifically, for the ACC task, the average improvement from the path information on all three test sets on the AbstRCT and PE datasets is 0.36 points and 1.2 points, respectively. This may be due to the model without path information already reaching high performance on this subtask. As for the ARI and ARC subtasks, generating path information results in improvements of 2.9 and 2.5 points on the PE dataset, which shows the positive impact of the path information and indicates that the "chain of thought" method is useful for the argument mining task. The path information is most important for the ARIC subtasks. Here, without the path information, the performance drops 3.1 points on average, possibly because the ARIC subtask is a combination of the ARI and ARC subtasks which both benefit from path information.

## 4.6 Impact of the Path

In Section 4.5, we only show the general impact of the path for the argument mining task by comparing model performance with and without path information. Here, we examine the effect of path information.

|      | 1     | 2     | 3     | 4     | all   |
|------|-------|-------|-------|-------|-------|
| ACC  | 83.19 | 70.49 | 38.09 | 6.25  | 70.61 |
| ARI  | 87.72 | 71.63 | 31.59 | 10.41 | 72.16 |
| ARC  | -     | 89.51 | 57.14 | 0     | 81.33 |

Table 7: Accuracy of the predicted path on the PE dataset. Here, *1*, *2*, *3* and *4* refer to the length of the path; *all* denotes the accuracy on all lengths.

First, we calculate the accuracy of the predicted path. Here, only exactly matching true and predicted paths are treated as a correct instance. We not only report the overall accuracy, but also the accuracy according to the length of the path. It is clear in Table 7 and Table 9 that an increasing path length leads to significant drops in prediction accuracy. This is in line with our hypothesis that longer paths are harder to learn for the model. Another interesting phenomenon is that the overall accuracy on the ARI and ARIC subtasks is higher than that on the ACC subtask. One possible reason is that the path prediction subtask requires the understanding of ARs, and thus might benefit from the ARI and ARIC subtasks.

Furthermore, we explore whether correctly predicting the path indeed improves the overall task performance. Therefore, we compare the results of MRC_GEN with MRC_GEN(-path) when the path is predicted correctly or wrongly, respectively. Specifically, we first run the MRC_GEN model and then split the test set into parts where the predicted path is correct and incorrect. Then, we report the performance difference between the two models on these two subsets. The results are shown in Table 10 and Table 11. On the PE dataset, it is clear that with the path information, the performance of our model drops when the predicted path is wrong, while it increases when the predicted path is correct. This means that the performance truly improves because of correctly predicted paths.

More interestingly, however, for the AbstRCT dataset, the performance increases on most of the subtasks on all three test sets regardless of the correctness of the predicted path. To investigate this issue, we manually analyse the cases

where MRC_GEN(-path) predicts the wrong answer while MRC_GEN predicts the correct answer even though the path is predicted wrongly. We find two main behaviours. We call the first one *path extension*, where the predicted path is an extension of the ground truth path. It usually occurs on sub-graphs with a smaller number of nodes, when an argumentative graph consists of two unconnected sub-graphs. Because in most cases, the smaller sub-graphs do not contain the full "chain of thought", the model may learn it from the biggest sub-graph to have a more general understanding of an argument. One example can be seen in Example 1 of Table 8. There are two unconnected sub-graphs, the smaller one containing AC3 and AC7. To understand the full "chain of thought", AC6 is wrongly predicted as the predecessor of AC7, while the model correctly predicts the answer. The second behaviour, which we refer to as *claim replacement*, occurs when one claim is exchanged with another one with similar high-level semantics to conclude the whole paper. As shown in Table 8, the model wrongly predicted the path as '<none> <AC9> → <support> <AC6>' given Example 2 due to the similarity of AC8 and AC9. Seemingly, in these two situations, a slightly wrong path is also beneficial to the model. We sampled 30 examples where the path was predicted wrongly and manually analysed them. We found that 16 were instances of path extension whereas 6 were instances of claim replacement.

## 4.7 Tree vs. Non-tree Argument Structure

As we mentioned in Section 4.1, the argumentative graphs in the two datasets exhibit different structures (non-tree structure for the AbstRCT dataset vs. tree structure for the PE dataset). In this section, we will discuss the impact of the difference in structure.

Intuitively, the graph structure is more random compared with the tree structure. That might be the reason that the accuracy of the path prediction on the PE dataset (from 70.61 to 81.33 on all subtasks) is higher than that on the AbstRCT dataset (from 60.34 to 66.44) as shown in Table 7 and Table 9.

Another conclusion is that the path information might be more useful on the graph-based dataset. From Table 10 and Table 11 we can see that even when the path information is predicted wrongly, the model could still improve the performance. See Example 1 in Table 7. The argumentative graph

|  | Example 1 | Example 2 |
|---|---|---|
| Text | <AC3> Kaplan-Meier estimates showed a trend in overall survival favoring epoetin alfa (P =.13, log-rank test), <non_AC> ... < AC6> Epoetin alfa safely and effectively ameliorates anemia and significantly improves QOL in cancer patients receiving nonplatinum chemotherapy.< AC7 > Encouraging results regarding increased survival warrant another trial designed to confirm these findings. | ...<AC6> Hepatic glucose production decreased after rapamycin pre-treatment (- 1.1 ± 1.1 mg/kg/min, p = 0.04) and after ITx (- 1.6 ± 0.6 mg/kg/min, p = 0.015), <non_AC>... <AC8> Rapamycin pre-treatment before ITx succeeds in reducing insulin requirement, enhancing hepatic insulin sensitivity. <AC9> This treatment may improve short-term ITx outcomes, possibly in selected patients with T1DM complicated by insulin resistance. |
| Rel | (<AC1> sup <AC6>), (<AC2> sup <AC6>), (<AC4> sup <AC6>), (<AC5> sup <AC6>), (<AC3> sup <AC7>) | (<AC2> sup <AC8>), (<AC3> sup <AC8>), (<AC6> sup <AC8>) |
| TP | '<none> <AC7>' | '<none> <AC8>' → '<support> <AC6>' |
| PP | '<none> <AC6>' → '<support> <AC7>' | '<none> <AC9>' → '<support> <AC6>' |

Table 8: Two examples where MRC_GEN(-path) predicts the wrong answer while MRC_GEN gets the correct answer even though the path is predicted wrongly. Rel denotes the relations in the given argumentative text. TP denotes "true path" and PP represents "predicted path".

|  |  | 1 | 2 | 3 | 4 | all |
|---|---|---|---|---|---|---|
| ACC | NEO | 67.60 | 69.81 | 21.49 | 9.52 | 60.73 |
|  | GLA | 62.24 | 66.22 | 5.3 | - | 61.89 |
|  | MIX | 60.46 | 67.29 | 24.11 | 22.22 | 60.44 |
| ARIC | NEO | 69.92 | 64.09 | 27.41 | 19.04 | 60.34 |
|  | GLA | 66.53 | 66.11 | 13.33 | - | 64.08 |
|  | MIX | 67.13 | 69.30 | 48.93 | 22.22 | 66.44 |

Table 9: Accuracy of the predicted path on the AbstRCT dataset. Here, *1*, *2*, *3* and *4* refer to the length of the path; *all* denotes the accuracy on all lengths.

|  | ACC | ARI | ARC |
|---|---|---|---|
| Wrong_path | -9.22 | -4.42 | -4.08 |
| Correct_path | +4.13 | +6.09 | +3.97 |

Table 10: The difference between MRC_GEN and MRC_GEN(-path) on the PE dataset when the path is predicted correctly or wrongly. Here, (Wrong/Correct)_path means that the path is predicted wrongly/correctly. Positive values mean that the path information improves the performance.

|  |  | ACC | ARIC |
|---|---|---|---|
| NEO | Wrong_path | +1.07 | +2.86 |
|  | Correct_path | +0.30 | +2.45 |
| GLA | Wrong_path | -1.49 | +11.52 |
|  | Correct_path | +0.66 | +4.75 |
| MIX | Wrong_path | -1.49 | +3.67 |
|  | Correct_path | +0.81 | +3.31 |

Table 11: The difference between MRC_GEN and MRC_GEN(-path) on the AbstRCT dataset when the path is predicted correctly or wrongly.

consists of two unconnected subgraphs. The first one includes five ACs with AC1, AC2, AC4 and AC5 supporting AC6. The second one contains only two ACs (AC3 supports AC7). When AC7 is used as a query, the true path is '<none> <AC7>'. However, the model predicts a wrong path '<none> <AC6>'→ '<support> <AC7>' that includes the information of AC6 and correctly predicts the relation between AC3 and AC7. It is clear that AC7 is a more general claim which can be used in many papers while AC6 and AC3 share some important information that is specific in this paper, such as *epoetin alfa*. From this point of view, AC6 can be regarded as a piece of implicit information for the relation between AC3 and AC7. Therefore, the path extension is helpful for the non-tree structures because some sub-graphs may not contain enough information. Meanwhile, for the tree structures, all the relations are connected explicitly. Thus, the model will surf from the wrongly predicted paths significantly.

## 5 Conclusion

In this paper, we cast the argument mining task as a multi-hop generative MRC task, which provides us with a means to leverage the "chain of thought" of an argument in a generative manner for the argument mining task. In addition, we also introduce a two-direction method to alleviate the order bias of the output sequences of our model. The extensive experimental results and detailed analysis demonstrate the positive impact of the "chain of thought". One direction for future work is quantifying the error of the path prediction to develop a reward model and train the generative model on that using proximal policy optimization.

## Limitations

In this paper, we show that the "chain of thought" information can improve the performance of argument mining models. However, such information may be not useful in user-generated arguments drawn from other domains such as social media, i.e., online forums, where the "chain of thought" can be more random compared with scientific abstracts and student essays. Another shortcoming is that we use only path information as the chain of thought. However, this captures only part of the semantics since argument structure is usually a graph rather than a single path.

## Acknowledgement

This work is supported by the computational shared facility at the University of Manchester and the project JPNP20006 from New Energy and Industrial Technology Development Organization (NEDO).

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

## A Multi-task Training

We also try to use a multi-task training paradigm. As mentioned in Section 3, we use the same query token <query> for all the subtasks. However, it is not suitable for the multi-task training setting because given the same query and the context, the model needs to predict different answers according to the different subtasks. In this setting, we need to distinguish queries for different subtasks. Concretely, we use <q_acc>, <q_ari>, <q_arc> and <q_aric> to represent the queries for the ACC, ARI, ARC and ARIC subtasks.

The loss function for each subtask is the same as the single subtask setting, and the final loss for both subtasks is shown as below:

$$\mathcal{L}_{AbstRCT} = \mathcal{L}_{ACC} + \mathcal{L}_{ARIC} \qquad (6)$$

$$\mathcal{L}_{PE} = \mathcal{L}_{ACC} + \mathcal{L}_{ARI} + \mathcal{L}_{ARC} \qquad (7)$$

We report the results in Table 12 and Table 13. It can be seen that the results of the multi-task training for most subtasks on both datasets are lower than the single task training setting. For the AbstRCT dataset, the reason may be that the difference between the best learning rate for the ACC (3e-5) and ARIC (8e-5) subtasks is large. Meanwhile, on the PE dataset, our results are in line with previous works (Bao et al., 2021; Kuribayashi et al., 2019) which found that multi-task training for all the three subtasks decreases the performance. One possible factor is the weight of each subtask. In our paper, we use 1:1:1 when there are three subtasks (ACC, ARI and ARIC) and 1:1 when there are two subtasks (ACC and ARIC). Fine-tuning this hyper-parameter might improve the performance.

## B Graph as Reasoning Path

We also conducted some initial experiments to leverage the whole graph information instead of

|      |     | Single | Multi |
|------|-----|--------|-------|
| ACC  | NEO | 92.76  | **92.83** |
|      | GLA | 92.62  | **93.03** |
|      | MIX | 93.97  | **94.26** |
| ARIC | NEO | **74.97** | 74.35 |
|      | GLA | **74.28** | 72.06 |
|      | MIX | **73.87** | 73.14 |

Table 12: Results of multi-task training and single task training on the AbstRCT dataset.

|        | ACC  | ARI  | ARC  |
|--------|------|------|------|
| Single | **89.2** | **82.7** | **78.2** |
| Multi  | 87.5 | 80.2 | 72.4 |

Table 13: Results of multi-task training and single task training on the PE dataset.

the path information since the path information can only include part of the argument structure information. Since our model is a BART-Base model and the output is only a sequence but not a graph, we need to transfer the graph representation into a sequence. We propose two ways to do this. Due to the nature of argumentation, argumentative graphs are all directed acyclic graphs. Therefore, we use a graph traversal algorithm, namely topological sorting, to represent a graph as a sequence. Given an argumentative graph $\mathcal{G} = \{\mathcal{V}, \mathcal{E}\}$, the algorithm returns a sequence of nodes:

$$< AC_1 >< AC_2 > ... < AC_n > \qquad (8)$$

In order to show the hierarchical structure of the graph, we use '|' as a separator between different layers.

$$G = < AC_{l_1^1} >< AC_{l_1^2} > | < AC_{l_2^1} > ... \qquad (9)$$

In addition, we also leverage the adjacency matrix to represent the argumentative graph inspired by (Guo et al., 2022; Bao et al., 2022). Each edge is represented by the start and end pairs of the edge, and '|' is used to distinguish different edges:

$$\begin{aligned} G = &< AC_1 >< AC_2 > |... \\ &| < AC_i >< AC_j > |... \qquad (10) \\ &| < AC_{n-1} >< AC_n > \end{aligned}$$

From Table 14 we find that the models using the graph information instead of the path information work poorly even when compared with the case where there is no further information included. It

|      |      | no_path | graph_adj | graph_topo |
|------|------|---------|-----------|------------|
| ACC  | NEO  | 92.29   | 90.14     | **92.70**  |
|      | GLA  | 92.27   | 91.92     | **93.02**  |
|      | MIX  | 93.72   | 91.20     | **94.14**  |
| ARIC | NEO  | **72.47** | 67.79   | 64.03      |
|      | GLA  | **70.10** | 64.99   | 62.74      |
|      | MIX  | **71.20** | 67.45   | 64.75      |

Table 14: Results of leveraging the whole graph information on the AbstRCT dataset.

seems that the topological sorting representation is helpful on the ACC subtask; however, it decreases the performance on the ARIC subtask significantly.

In general, we believe that the reason why including the whole argument graph instead of the reasoning path harms performance is that the sequence-to-sequence model is not strong enough to learn the whole graph information. The accuracy (based on exact match) of the predicted graph is too low (5.86% on the AbstRCT dataset). We think that there are mainly two reasons. First, the number of tokens for the representation of a whole graph (from 5 tokens to 26 tokens) is much longer than that of a path (from 1 token to 4 tokens). As shown in Table 7 and Table 9, the accuracy of the path prediction drops significantly with the increase of the path length. When the path length is 4, the accuracy can be only 9.52%. Another reason is that the semantics of a path is in a sequential order which is consistent with the learning method of pre-trained language models, while the graph structure is more complex and is not easy to learn in a sequential order.

## C Hyperparameter Analysis

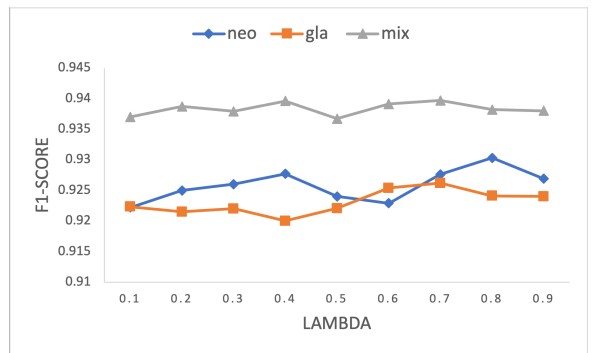

Figure 2: The performance on the ACC subtask based on the AbstRCT dataset using different values of $\lambda$.

In this section, we investigate the effect of the loss weight $\lambda$. The value of $\lambda$ is set between 0.1

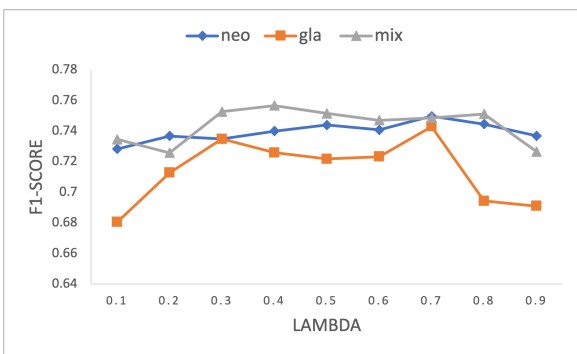

Figure 3: The performance on the ARIC subtask based on the AbstRCT dataset using different values of $\lambda$.

and 0.9. As mentioned in Section 3.4, the higher $\lambda$ is, the less attention is paid to the path information. Figure 2 and Figure 3 show the impact of $\lambda$ on the AbstRCT dataset. It is clear that when $\lambda$ is too small, the model cannot learn the answer part of the output sequence well, which causes a lower performance. However, if the $\lambda$ is too large, the model mainly concentrates on the answer part and the path information is not fully understood by the model.