# OpenReview forum: "Argument mining as a multi-hop generative machine reading comprehension task"
_EMNLP/2023/Conference — EMNLP 2023 Findings_

### Official Review · Reviewer_As1i · 2023-08-03

**Soundness:** 4

**Excitement:**

3: Ambivalent: It has merits (e.g., it reports state-of-the-art results, the idea is nice), but there are key weaknesses (e.g., it describes incremental work), and it can significantly benefit from another round of revision. However, I won't object to accepting it if my co-reviewers champion it.

**Paper Topic And Main Contributions:**

This paper addresses the argument mining task by transferring it to a multi-hop reading comprehension task (MRC). To this end, the authors process the instances into a quadruple of <query, context, answer, path> based on the task specification. For example, for argument unit classification, the query is the component to be classified, context is the whole essay, the answer is the label of the component (claim, main claim, etc.), and the path is a sequence of components starting from the main component to the query component, representing a chain of thoughts. Given this processed dataset, the authors train a BART model on jointly predicting the path and the answer.

In a set of experiments, the authors compare their approach to several baselines on two datasets. The results demonstrate a degree of boost gained from the implemented approach.

The paper is well-written and easy to follow in most of its parts. The experiments conducted, including the selected baselines, are adequate. However, details on the training process and evaluation measures are not clear.

UPDATE: After the authors response that clarified my concerns, I decided to increase my scores.

**Questions For The Authors:**

- Could you provide more details on the training process? How many epochs? Any fine-tuned hyperparameters? Did you perform cross-fold validation? That can make your results stronger and more reliable.

- Did you think of implementing your idea as a prompt for large language models? It would be nice to see how that works

**Reasons To Accept:**

- Converting the task into a multi-hop reading comprehension task is interesting and well-motivated.

- The experiments are sound.

- The paper is well-written in most of its parts.

**Reasons To Reject:**

- The results don't seem very reliable to me, especially since it is not clear whether there was any cross-fold validation performed or significance test

**Reproducibility:**

2: Would be hard pressed to reproduce the results. The contribution depends on data that are simply not available outside the author's institution or consortium; not enough details are provided.

**Reviewer Confidence:**

3: Pretty sure, but there's a chance I missed something. Although I have a good feel for this area in general, I did not carefully check the paper's details, e.g., the math, experimental design, or novelty.

---

> ### Author Rebuttal · Authors · 2023-08-28
>
> Thank you very much for your review. We read it carefully and try our best to solve your concerns. The details are shown below.
>
> **Response to "Reasons To Reject"**:
>
> As mentioned in line 336 to line 343 (“All results on both datasets are averaged scores of three different random seeds and are reported as macro-averaged F1 scores. For the AbstRCT dataset, we use the same train-development-test split as Si et al. (2022). For the PE dataset, we keep the same train-test split and randomly select 10% of the training set for validation, like Bao et al. (2021) do.”), we followed the same evaluation method as previous models so that we could get a fair comparison between our model and the baseline models. Therefore, we did not use cross-fold validation in the evaluation.
>
> [1] Jianzhu Bao, Yuhang He, Yang Sun, Bin Liang, Jiachen Du, Bing Qin, Min Yang, and Ruifeng Xu. 2022. A generative model for end-to-end argument mining with reconstructed positional encoding and constrained pointer mechanism. In Proceedings of the 2022 Conference on Empirical Methods in Natural Language Processing, pages 10437–10449, AbuDhabi, United Arab Emirates. Association for Computational Linguistics.
>
> [2] Jiasheng Si, Liu Sun, Deyu Zhou, Jie Ren, and Lin Li. 2022. Biomedical argument mining based on sequential multi-task learning. IEEE/ACM Transactions on Computational Biology and Bioinformatics.
>
> As for the significance test, we agree that it is a good way to improve the reliability of our results. Thus, we conducted a significance test and the results are shown below.
>
>
> |        |         |  ACC   |           |          |  ARIC |           |
> |  ----  | ----  |  ----  | ----  | ----  |  ----  | ----  |
> |   | Neo  |  Gla  | Mix  |Neo  |  Gla  | Mix  |
> | ResArg(avg) |  86.18 | 85.53|  86.74 | 59.15 | 57.23|  60.31|
> | ResArg(Ensemble)  |86.38 |87.13 |87.59 |63.16 |61.86 |68.35|
> | ResAttArg(avg)  | 86.19| 86.26| 87.51| 66.49| 62.68| 63.47 |
> | ResAttArg(Ensemble)  | 87.87| 87.71| 89.70| 70.92| 68.40| 67.66|
> | SeqMT  | 91.89| 92.35| 92.21| 71.24| 73.27| 72.71|
> | MRC_GEN  | **92.76***| **92.62***| **93.97***| **74.97***| **74.28***| **73.87***|
> |        |         |    |           |          |  |           |
>
> Table 1: Overall results on the AbstRCT dataset. Here, Neo, Gla and Mix correspond to the results achieved for the neoplasm, glaucoma and mixed test sets, respectively. The highest scores are emboldened. * indicates statistically significant improvements over the baselines with our model (*: by pair-wise t-test with p < 0.05)
>
> |        |         |  ACC   |           |          |  |   ARI |          |  |   ARC |   |
> |  ----  | ----  |  ----  | ----  | ----  |  ----  | ----  | ----  | ----  |  ----  | ----  |
> |   | Macro |MC |Claim |Premise |Macro |Rel| No-Rel| Macro| Support |Attack |
> | Joint-ILP |82.6 |89.1 |68.2 |90.3| 75.1 |58.5 |91.8 |68.0| 94.7 |41.3 |
> | St-SVM-full |77.6 |78.2| 64.5 |90.2 |- |60.1| -| -| - |-|
> | Joint-PN |84.9 |89.4| 73.2 |92.1 |76.7| 60.8 |92.5| - |-| -|
> | Span-LSTM |87.3| - |- |- |81.1| - |- |79.0 |96.8 |**61.1** |
> | BERT-Trans |88.4 |93.2| 78.8| 93.1| 82.5 |70.6| 94.3| **81.0**| - |-|
> | MRC_GEN |**89.2*** |**94.8***| **79.6*** |**93.2** |**82.7***| **70.9*** |**94.4**| 78.2*| **97.7***| 58.9*|
> |        |         |    |           |          |  |           |  | |   ||
>
> Table 2: Overall results on the PE dataset.* indicates statistically significant difference between the baselines with
> our model (*: by pair-wise t-test with p < 0.05)
>
> Please note that we cannot conduct the significance test to compare our model with SeqMT on the AbstRCT dataset since their code is not available and we cannot get enough information from their paper for a significance test. Therefore, our significance test is based on other models.  If our paper is accepted, we will add the results of the significance test into our paper.
>
> **Response to "Questions For The Authors"**:
>
> **Q.a** As mentioned in line 347, the details of the hyper-parameters are shown in Appendix B due to the limitation of the space. We agree that this type of information is important. If our paper is accepted, we will move it to the main context so that the readers can find it easily.
>
> **Q.b** Thanks for your suggestion. We agree that it is worth trying our idea in large language models. We proposed a generative model to solve the argument mining task mainly because the large language model era is coming. We will try it in the future.

---

### Official Review · Reviewer_SLiR · 2023-08-03

**Soundness:** 4

**Excitement:**

4: Strong: This paper deepens the understanding of some phenomenon or lowers the barriers to an existing research direction.

**Paper Topic And Main Contributions:**

This paper aims to transfer the argument mining task into a multi-hop reading comprehension task adopting the innovative concept of “chain of thought”.
The contribution of this paper involves the novel utilization of the concept of “chain of thought”, coupled with a comprehensive evaluation process.



**Reasons To Accept:**

The results support the efficacy of the model introduced by the authors for this particular task.



**Reasons To Reject:**

One potential constraint of this study lies in its focus on a single path, without considering the inherent variability of the "chain of thought".



**Reproducibility:**

4: Could mostly reproduce the results, but there may be some variation because of sample variance or minor variations in their interpretation of the protocol or method.

**Reviewer Confidence:**

2: Willing to defend my evaluation, but it is fairly likely that I missed some details, didn't understand some central points, or can't be sure about the novelty of the work.

---

> ### Author Rebuttal · Authors · 2023-08-28
>
> Thank you very much for your review. We read it carefully and try our best to solve your concerns. The details are shown below.
>
> **Response to "Reasons To Reject"**:
>
> Thank you very much for your comments! It is a good suggestion to consider the inherent variability of the "chain of thought" instead of only using a single path. We will consider that for our future work.

---

### Official Review · Reviewer_g9cj · 2023-08-11

**Soundness:** 3

**Excitement:**

4: Strong: This paper deepens the understanding of some phenomenon or lowers the barriers to an existing research direction.

**Paper Topic And Main Contributions:**

Viewing the argument structure within a text as akin to a "chain of thought", the authors propose to formulate the argument mining (AM) task as a multi-hop machine reading comprehension (MRC) task, allowing the model to learn the argument structure as a sequence of reasoning steps leading to a conclusion. By converting AM into a generative multi-hop MRC task that takes a query and context as its input and outputs an answer plus the predicted path, they enable the model to learn the “chain of thought” and extract the argument structure simultaneously. Additionally, the authors use a bi-directional method to mitigate the order-induced bias. The proposed method and its different components are tested through experimentation on two benchmark datasets. The experimental results show that this method achieves the SOTA on all or most tasks for both datasets, which speaks to the utility of the "chain of thought" approach and the corresponding MRC method for argument mining.

**Questions For The Authors:**

(Q.a) What are the implications of the difference between PE and AbstRCT for the performance of the method on each task, especially in light of the ablation study (e.g. table 9 VS table 10)?

(Q.b) The authors claim on line 428 that “The warm start method improves the performance in most cases”. This is while the performance on ACC on all three test sets in AbstRCT is worse with a warm start. This does not seem to be a random observation, and the reason does not seem clear, especially since path prediction for ARIC does not seem much better than ACC (table 8), i.e., worse path prediction might not be a cause of this difference. What is causing this then? (The cause could be unknown to the authors, but a discussion pointing this out seems appropriate.)

(Q.c) Line 525 reports some statistics on occurrence of path extension and claim replacement: 16 and 6 occurrences (respectively) out of a sample of 30 observations. Are these 30 examples instances of wrong predicted path but correct answer? If so, is there any other feature for the remaining 8 instances that stands out?

(Q.d) (optional) Why and in what ways does including the whole argument graph instead of the reasoning path hurt the performance? This is to some extent discussed in Appendix D, but a brief mention of this in the main text, beyond a footnote, could be helpful.

(Q.e) (optional) Are the baseline models trained via teacher forcing? If not, how do their performances compare with the proposed method without teacher forcing? Or how do they perform if they are trained via teacher forcing?


**Reasons To Accept:**

(S.a) The paper proposes a novel, intuitive, and clearly described method to take advantage of the “chain of thought” structure for AM.

(S.b) The experimental results suggest that this approach, formulated as an MRC task and implemented through the proposed method, could help to learn useful information from the reasoning path, improving the AM performance.

(S.c) The ablation study appropriately tests the contribution of each component and, as highlighted by the authors, the experiments offer useful insight pointing to directions for future research.

(S.d) Overall, the experiments alongside the core idea of the "chain of thought" structure behind utilizing MRC, open paths with potentially vast opportunities for future research.


**Reasons To Reject:**

(W.a) While the experiments are overall well-designed, the results deserve further discussion, investigation, or interpretation. The following are some examples:

- The authors find that including the full argument graph instead of only the path hurts the performance, but this, not necessarily intuitive observation, remains as a footnote referring to an appendix.
- The authors find that training all subtasks on all the dataset at once (instead of teacher forcing) hurts the performance.
- There is a clear difference between the two datasets with regards to the impact of warm start. There is no discussion on why PE seems to benefit from warm start on all tasks, while warm start is clearly hurting the ACC task on AbstRCT.

(W.b) Related to the last point, the differences between the two datasets are underexplored. For example, the authors mention that the argument graph is a tree in PE, but not in AbstRCT. This is a potentially important difference, given the core idea of this method. In light of the differences in performance, this deserves more discussion.

(W.c) Speculative or equivocal language could be replaced with more quantitative information. For example, the authors remark on line 504 that path extension “usually occurs on subgraphs with a smaller number of nodes”. This sentence does not give a reader who is unfamiliar with the data or the task a clear idea of what is considered “small” and how often is considered “usually”. This could be clarified with some statistics on the distributions of graph size and occurrence of path extension.


**Reproducibility:**

4: Could mostly reproduce the results, but there may be some variation because of sample variance or minor variations in their interpretation of the protocol or method.

**Reviewer Confidence:**

2: Willing to defend my evaluation, but it is fairly likely that I missed some details, didn't understand some central points, or can't be sure about the novelty of the work.

---

> ### Author Rebuttal · Authors · 2023-08-28
>
> Thank you very much for your review. We read it carefully and try our best to solve your concerns. The details are shown below. Please note that we use the same serial number as you used to reply to your comments (i.e., Q.a means the answer to your question Q.a).
>
> **Response to"Reasons To Reject"**:
>
> **W.a.1**. Due to the limitation of the space, we moved this part into the Appendix. We agree that it is important. Therefore, if our paper is accepted, we will move this part into the main text.
>
> **W.a.2**. That is really one phenomenon that confuses us. However, we are not the only ones met this problem. As we mentioned in Appendix C, previous papers (Bao et al., 2021; Kuribayashi et al., 2019) already show that their models suffer from joint learning of all three subtasks. Another successful multi-task method, SeqMT (Si et al., 2022), implements a multi-task learning framework that leverages the sequential dependency between the ACC and ARIC subtasks by transferring the representation of the input and output of the ACC subtask to the ARIC subtask. SeqMT shows the positive impact of joint learning by leveraging the sequential dependency between the ACC and ARIC subtasks. Based on these three papers, we think that a simple joint learning method is not enough to learn the complex relation between the subtasks of argument mining and some better methods are needed to model this complex relation. Another possible impact factor is the weight of each subtask. In our paper, we use 1:1:1 when there are three subtasks (ACC, ARI and ARIC) and 1:1 when there are two subtasks (ACC and ARIC). Fine-tuning this hyper-parameter might improve the performance. We will add these analyses to our paper if it is accepted so that some other researchers could also join to find the reasons or develop a better joint learning method.
>
> [1] Jianzhu Bao, Yuhang He, Yang Sun, Bin Liang, Jiachen Du, Bing Qin, Min Yang, and Ruifeng Xu. 2022. A generative model for end-to-end argument mining with reconstructed positional encoding and constrained pointer mechanism. In Proceedings of the 2022 Conference on Empirical Methods in Natural Language Processing, pages 10437–10449, AbuDhabi, United Arab Emirates. Association for Computational Linguistics.
>
> [2] Tatsuki Kuribayashi, Hiroki Ouchi, Naoya Inoue, Paul Reisert, Toshinori Miyoshi, Jun Suzuki, and Kentaro Inui. 2019. An empirical study of span representations in argumentation structure parsing. In Proceedings of the 57th Annual Meeting of the Association for Computational Linguistics, pages 4691–4698, Florence, Italy. Association for Computational Linguistics.
>
> [3] Jiasheng Si, Liu Sun, Deyu Zhou, Jie Ren, and Lin Li. 2022. Biomedical argument mining based on sequential multi-task learning. IEEE/ACM Transactions on Computational Biology and Bioinformatics.
>
> **W.a.3**. It is a good question. We are also interested in why the warm start method does not work on the ACC subtask on the AbstRCT dataset. However, since we directly used the warm start method and it is not our contribution and concentration, we can just give some simple reasons based on our experimental results. We guess one possible reason is that fine-tuning hyper-parameters could have a similar effect with the warm start method as the warm start could only provide better start points for the specific tokens <ACi> but not better final representations.
>
> We got this conclusion from our experimental process. In the beginning, we used a set of common hyper-parameters to test the performance of our model and we compared the performance of the two models with and without warm-start, we found that warm-start could improve the performance on all the subtasks. Then we started fine-tuning the hyper-parameters and chose the ones that achieved the best performance. Then we did an ablation study based on these hyper-parameters, from which we found that the warm start method hurt the performance on the ACC subtask when the AbstRCT dataset is used. The best learning rate for the ACC subtask on the AbstRCT dataset is 3e-5. When we use 2e-5 as the learning rate, the warm start method works (the F1-scores are 92.50, 91.84 and 93.71 when warm start is used and are 92.21, 91.66 and 93.59 without warm start).
>
> From an intuitive point of view, each AC token needs to include two types of information, the location of the AC and the content of the AC. The warm start method only includes location information since we use the embedding of the numbers as a start point for each AC token, while the model also needs to learn that this token is about the content of the AC. However, overlearning the representation from the warm start method may also decrease the performance. Therefore, good hyper-parameters are also important to let the model learn a balanced representation of these two types of information. So we think the possible reason is that the hyper-parameters for the ACC subtask on the PE dataset are good enough to learn a balanced representation.
>
> **W.b**. We agree that discussing the differences between different types of argument structures (graph or tree) is very useful. The reason we did not provide this type of information is that it is impossible to give a direct comparison as the differences between the two datasets are multiple. To be specific, they are in different domains(the PE dataset is the general domain and the AbstRCT dataset is the biomedical domain), different genres (student essay and scientific paper) and different lengths (the average length of the texts in the AbstRCT dataset is much longer than that in the PE dataset).
>
> However, we can still obtain something based on the differences in the performances of the two datasets. Intuitively, the graph structure is more random compared with the tree structure. That might be the reason that the accuracy of the path prediction on the PE dataset (from 70.61 to 81.33 on all subtasks) is higher than that on the AbstRCT dataset (from 60.34 to 66.44) as shown in Table 6 and Table 8.
> Another conclusion is that the path information might be more useful on the graph-based dataset. From Table 9 and Table 10 we can see that even when the path information is predicted wrongly, the model could still improve the performance. See Example 1 in Table 7. The argumentative graph consists of two unconnected subgraphs. The first one includes 5 ACs with AC1, AC2, AC4 and AC5 supporting AC6. The second one only contains two ACs (AC3 supports AC7). When AC7 is used as a query, the true path is ’<none> <AC7>’.  However, the model predicts a wrong path “<none> <AC6>’→ ’<support> <AC7>” that includes the information of AC6 and correctly predicts the relation between AC3 and AC7.  This phenomenon is called path extension in Line 503. The content of these three ACs are shown below:
>
> <AC3>: *Kaplan-Meier estimates showed a trend in overall survival favoring epoetin alfa (P =.13, log-rank test),*
>
> <AC6>: *Epoetin alfa safely and effectively ameliorates anemia and significantly improves QOL in cancer patients receiving nonplatinum chemotherapy.*
>
> <AC7>: *Encouraging results regarding increased survival warrant another trial designed to confirm these findings.*
>
> It is obvious that AC7 is a more general claim which can be used in many papers while AC6 and AC3 share some important information that is specific in this paper, such as epoetin alfa. From this point of view, AC6 can be regarded as a piece of implicit information for the relation between AC3 and AC7. Therefore, the path extension is helpful for the graph structures because some sub-graphs may not contain enough information. While for the tree structures,  all the relations are connected explicitly. Thus, the model will surf from the wrongly predicted paths significantly.
>
> **W.c**. We accept this suggestion. We agree that quantitative information is more perspective than speculative or equivocal language. However, it is worth mentioning that “a smaller number of nodes” does not mean a range of node numbers here. As we mentioned in line 504 to line 507,  “It usually occurs on sub-graphs with a smaller number of nodes, when an argumentative graph consists of two unconnected sub-graphs.”  Here “sub-graphs with a smaller number of nodes” means the smaller sub-graph of the two unconnected sub-graphs. We will clarify it in the next version of our paper.
>
> **Response to "Questions For The Authors"**:
>
> **Q.a**. As this question is similar to W.b, please back to the answer W.b.
>
> **Q.b**. We agree that the accuracy of the path prediction is not the reason for that as the same contexts are used for the prediction of the path and only the queries are different for different tasks. For more details of the possible reason, please back to our answer W.a.3.
>
> **Q.c**. These 30 examples are instances where with a wrongly predicted path, the model corrects the answer compared with the model when path information is not included. For the other remaining instances, there are no obvious common features. It seems that the model predicted the correct answer even though the path was predicted wrongly.
>
> **Q.d**. In general, we think the reason why including the whole argument graph instead of the reasoning path hurt the performance is that the sequence-to-sequence model is not strong enough to learn the whole graph information.
>
> Our original idea is to use graph information as the representation of the chain of thought. However, as you can see in Appendix D, we tried different ways to get the graph representation but the performance was bad. The main reason we think is that the accuracy (exact match) of the predicted graph is too low (5.86% on the AbstRCT dataset). We think there are mainly two reasons. First, the number of tokens for the representation of a whole graph (from 5 tokens to 26 tokens) is much longer than that of a path (from 1 token to 4 tokens). As shown in Table 6 and Table 8, the accuracy of the path prediction drops significantly with the growth of the path length. When the path length is 4, the accuracy can be only 9.52%. Another reason is that the semantics of a path is in a sequence order which is consistent with the learning method of PLM, while the graph structure is more complex and is not easy to learn in sequence order.
>
> We will add the analysis above to our paper if it is accepted.
>
> **Q.e**. As none of the baselines use a generative model to solve the argument mining task, teacher forcing is unavailable for these models.

---

### Meta-Review · Area_Chair_if88 · 2023-09-19

**Recommendation:** 3

**Metareview:**

The reviewers agreed that the proposed method to take advantage of the “chain of thought” structure for Argument Mining is novel and sound. The reviewers highlighted some drawbacks in their reviews (i.e., the experiments and the results lack of key details like on the training process, and deserve a deeper investigation like for the issue of considering full argumentation graphs). The reviewers appreciated the author rebuttal which addressed most of the mentioned issues.

---

### Decision · Program_Chairs · 2023-10-07

**Decision:**

Accept-Findings

**Comment:**

The reviewers agreed that the proposed method to take advantage of the “chain of thought” structure for Argument Mining is novel and sound. The reviewers highlighted some drawbacks in their reviews (i.e., the experiments and the results lack of key details like on the training process, and deserve a deeper investigation like for the issue of considering full argumentation graphs). The reviewers appreciated the author rebuttal which addressed most of the mentioned issues.